# A New Method to Assess Thick, Hard Roof-Induced Rock Burst Risk Based on Mining Speed Effect on Key Energy Strata

**Wenlong Li, Shihao Tu \*, Hongsheng Tu, Xun Liu, Kaijun Miao, Hongbin Zhao, Jieyang Ma, Long Tang and Yan Li**

State Key Laboratory of Coal Resources and Safe Mining, School of Mines, China University of Mining and Technology, Xuzhou 221116, China
\* Correspondence: tsh@cumt.edu.cn; Tel.: +86-13952160512

**Abstract:** Roof-type rock burst (RTRB) frequently occurs in the hard, thick roof of working faces, which causes roadway failure, facility damage and even personnel casualties. Previous research results show that mining speed has obvious effects on the rock burst risk and many rock burst accidents are caused by an unreasonable mining speed. To provide a theoretical foundation for the determination of a reasonable mining speed in a specific working face subjected to RTRB, in this study, the key energy strata (KES) principle contraposing the RTRB was proposed, and the criterion of KES was determined by defining the energy release coefficient $k_c$. On this basis, the energy accumulation characteristics of coal and energy release of surrounding rock were analyzed using FLAC3D numerical simulation. Accordingly, to assess the rock burst risk considering the mining speed effect, a new method was proposed and a new energy index $\Phi_{vi}$ was defined to divide rock burst risk with different mining speeds into four grades. To validate the availability of the KES principle and the new assessment method, they were adopted in a thick, hard roof working face. The application results indicate that the mining speed of 3.6 m/d obtained by the method meets the demands of safe and high-efficiency production.

**Keywords:** thick, hard roof; roof-type rock burst; key energy strata principle; mining speed effect; evolution of energy accumulation/release; evaluation method of rock burst risk

## 1. Introduction

Rock burst has threatened the safety and efficiency of coal resource exploitation in many countries for many decades [1–3]. Due to the complex geological conditions and various mining technical conditions, rock burst in underground coal mining shows the characteristics of diverse disaster-causing factors [4–13]. Among them, roof-type rock burst (RTRB) occurs frequently during the mining process under thick, hard roofs [14–21].

Rock burst risk assessment forms a significant basis for the effective control of rock burst. When a thick, hard roof is broken, the radiated energy can be captured by the micro-seismic monitoring system, which is widely used as a tool to assess rock burst risk [22]. For instance, Cao et al. [23] investigated the microseismic multidimensional information for the identification of rock bursts and spatial–temporal prewarning in a specific coalface, which suffered high rock burst risk in a mining area near a large residual coal pillar, providing a reference for assessing rock burst risk and determining potential rock burst risk areas in coal mining. Zhu et al. [24] applied seismic velocity tomography to an island longwall panel, and a good hypothetical correlation between regions of high speed or velocity gradient anomalies, areas of high stress and rock burst risk was confirmed. Cai et al. [25] defined and simulated seismic energy in a numerical model, developed a damage mechanics model to correlate stress, strain, damage and seismic energy release and proposed a new index named "bursting strain energy" to quantitatively assess coal burst propensity. Chen et al. [26] proposed a seismic event impact hazard assessment

model based on multi-source parameters, and the application results show that the model can improve the efficiency of coal burst monitoring and early warning. Liu et al. [27] proposed a new method for impact coal pressure risk assessment based on seismic energy decay calculation, and proposed three seismic-based indicators considering static response and dynamic response and their superposition effects to assess impact coal pressure risk. Based on dynamic static energy principles, He et al. [28] studied the precursory rules of coal bumps combined with electromagnetic emission and microseismic monitoring, and established a coupling evaluation system according to the monitoring results of a coal mine. Li et al. [29] investigated the effect of the number of seismic data used for calculations and modifications to the arrangement of seismic monitoring transducers on the method of detecting earthquake probability. It can be concluded that these research results are mainly about risk assessment considering different geological factors. However, technical mining factors, especially mining speed, also have a significant influence on rock burst risk. Previous research results [30–34] show that an increase in mining speed contributes to an increase in energy accumulation and stress concentration in surrounding rock, resulting in an increase in rock burst risk. However, they only obtained a qualitative relationship between mining speed and rock burst risk, which is not enough to provide theoretical guidance for the optimization of mining speed in field mining activities. Therefore, it is necessary to explore a quantitative method of assessing rock burst considering the effect of mining speed.

Aiming at the above problem, this work presents a new way of evaluating rock burst risk considering the effect of mining speed based on the energy accumulation characteristics of coal and energy release of surrounding rock obtained by FLAC3D numerical simulation. On the basis of the minimum energy principle of rock mass dynamic failure and the dynamic and static load superposition principle of rock burst, a new energy index $\Phi_{vi}$ was defined to divide rock burst risk at different mining speeds into four grades. Consequently, a reasonable mining speed can be selected for a specific working face subjected to a rock burst threat. This method was adopted in the rock burst working face of Zhangshuanglou Coal Mine, Xuzhou, China, and the application results indicate that the mining speed of 3.6 m/d obtained using the method meets the demands of safe and high-efficiency production.

## 2. The Principle and Discriminant Criterion of KES for RTRB

### 2.1. The Principle of KES for RTRB

The key strata theory of rock strata control [35] is mainly used to analyze the movement law of rock strata. However, mining activities not only cause strata movement, but also cause the redistribution of stress and the evolution of energy accumulation and release in rock and coal mass, which are closely related to the breeding and development of rock burst.

Under the conditions of deep mining, the physical and mechanical properties of the overlying strata are diverse. The strata with large thickness and high strength are not easy to fracture naturally after coal seam mining, and easily form a suspended roof structure and accumulate elastic strain energy. When it breaks, the elastic strain energy is released and influences the coal in the mining space in the form of vibrational waves. The coal itself accumulates certain elastic strain energy under self-weight pressure, tectonic pressure and mining-induced pressure. A rock burst appears once the energy carried in the vibrational wave and accumulated by the coal exceeds a critical value.

However, it is found that not all mining tremors caused by the fracture of thick, hard roofs will induce rock burst. Cao [36] found that the mining seismic source generated by the break of low-level rock strata is closer to the stope and roadway surrounding rock; the shorter the propagation path of the vibration wave, the lower the energy attenuation degree, the stronger the dynamic load disturbance to coal mass, and the higher the rock burst risk. Jiang et al. [37] identifies that mining disturbances of a working face have a stronger immediate effect on the deformation, fracture and structural adjustment of low-level thick,

hard rock strata (such as the main roof). Accordingly, the mining seismic activity caused by the fracture of low level thick, hard strata has the same effect on the dynamic load disturbance to coal. Gao et al. [38] proposed that the rock burst induced by thick, hard roofs has a near–far-field effect. The intensity of rock burst is not only related to the energy release of the thick, hard roof, but also the horizon itself. Therefore, the energy distance ratio (the ratio of energy to distance) is defined to describe this effect.

To reveal the effect mechanism of rock strata on rock burst in coal measure strata, Chen et al. [39] performed uniaxial compression experiments on coal–rock combinations with different lithology and coal–rock ratios. The results indicate that the energy of the combination was mainly accumulated in the coal mass, based on which the concept of key strata for energy accumulation was proposed. When rock strata fracture, energy can be released and transferred to the coal mass. Due to the rock strata having different strengths and thicknesses and the energy released by its fracture having different attenuation degrees during its propagation in the rock mass medium, the degrees of rock strata affecting rock burst are different. On this basis, Mu [40] proposed the "key strata inducing rock burst" theory to quantificationally characterize the possibility of inducing rock burst by rock strata.

However, the above studies neglected the impact of the energy evolution of key strata on rock burst under different mining speeds. In order to reveal the role of each rock strata in rock burst breeding and development in coal measure strata, the key energy strata (KES) principle for rock burst is proposed by referring to the key strata theory of strata control. It is defined as follows: during mining of a working face threatened by RTRB, the rock strata that have the potential to induce rock burst can be referred to as KES.

### 2.2. The Discriminant Criterion of KES

Similar to structural key strata, physical and mechanical parameters such as rock thickness and strength can be used to distinguish KES. At the same time, the strata layer, energy level of mining seismicities caused by rock strata fracture and attenuation process during vibrational wave energy propagation are important factors.

The existing research conclusions show that the elastic energy released by rock strata at its initial fracture is greater than that at periodic fracture [41,42]. Therefore, the former is used as the main basis for determining the KES. Professor Aviershen [35] pointed out that the energy released by rock strata at its initial fracture is

$$U_{\mathrm{d}} = \frac{q^2 L^5 b}{576 E_r J} \tag{1}$$

where $q$ is load concentration above the rock strata, MPa; $L$ is the hanging length, m; $b$ is the tendency span, m; $E_{\mathrm{r}}$ is the elastic modulus, GPa; $J$ is the inertia moment, m$^3$.

The limit span of rock strata at its initial fracture [35] is

$$L = h\sqrt{2R_{\mathrm{T}}/q} \tag{2}$$

The inertia moment of rock strata is

$$J = h^3/12 \tag{3}$$

By substituting Equations (2) and (3) into Equation (1), the released energy at initial fracture of rock strata can be obtained as

$$U_{\mathrm{d}} = \frac{bh^2}{E_r}\sqrt{\frac{R_{\mathrm{T}}^5}{72q}} \tag{4}$$

where $h$ is the thickness of rock strata, m; $R_{\mathrm{T}}$ is the tensile strength of rock strata, MPa.

Due to the layered and heterogeneous characteristics of coal measure strata, vibrational wave energy usually undergoes geometric attenuation and inherent attenuation during

propagation in coal and rock mass [43]. The attenuated seismic wave energy $U_{dh}$ can be expressed as

$$U_{\text{dh}} = U_{\text{d}}H^{-2}e^{-\eta H} = \frac{bh^2}{E_r}\sqrt{\frac{R_T^5}{72q}}H^{-2}e^{-\eta H} \tag{5}$$

where $H$ is the propagation distance of the vibrational wave when rock strata fracture, which is set as the vertical distance between the middle line of each rock strata and coal seam, m; $\eta$ is the energy attenuation coefficient, which is related to the energy level of the mining earthquake source and rock properties.

Based on an SOS microseismic system, Liu et al. [43] monitored and analyzed the attenuation law of vibrational wave energy with different source energy levels. It was found that the energy attenuation coefficient decreases when the source energy level increases. In this paper, energy attenuation coefficients with different source energy levels are selected according to Table 1.

**Table 1.** Energy attenuation coefficient with different source levels [43].

| Range of Energy Level | Energy Attenuation Coefficient $\eta$ |
|---|---|
| $10^2 \sim 10^3$ J | $1.3968 \times 10^{-3}$ |
| $10^3 \sim 10^4$ J | $1.2046 \times 10^{-3}$ |
| $10^4 \sim 10^5$ J | $9.1698 \times 10^{-4}$ |
| $10^5 \sim 10^6$ J | $7.2674 \times 10^{-4}$ |
| $10^6 \sim 10^7$ J | $5.5714 \times 10^{-4}$ |

Normally, the minimum energy inducing rock burst is $1 \times 10^4$ J. Therefore, $1 \times 10^4$ J is set as a critical value of vibrational wave energy at the initial fracture of rock strata when it propagates to the coal seam, and the energy release coefficient is set as $k_c$ (calculated by $U_{dh}/U_c$). When $k_c \geq 1$, the related rock strata are judged as KES.

### 2.3. Engineering Example Analysis

The 23,908 longwall mining face of Zhangshuanglou Coal Mine is taken as a case study object. The eastern part of the longwall mining face has protective pillars comprising the haulage entry, the southern part is the gob of the 93,606 longwall mining face, and the western and northern parts are solid coal (unmined area), as shown in Figure 1. The strike length of the 23,908 working face is 588 m, and the inclined width is 194 m. As of 26 April 2022, the average advancing distance of the 23,908 working face is 110 m. The average buried depth of the working face is 1103.5 m. The thickness of the 9# coal seam is 0.1~4.1 m with an average thickness of 2.1 m. The dip angle is 22°~26° with an average dip angle of 24°.

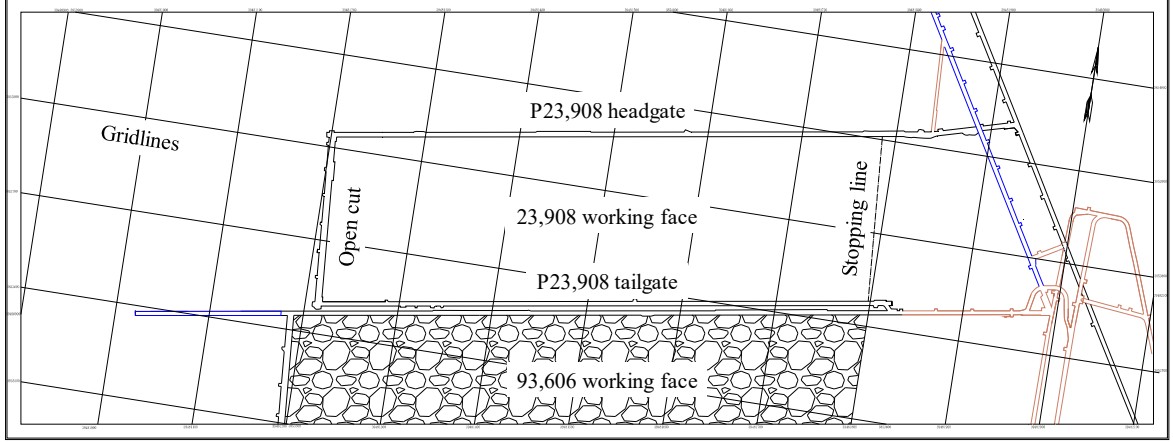

**Figure 1.** The layout of 23,908 working face.

There is a mudstone pseudo-roof locally development zone in the 23,908 working face. The thickness of the pseudo-roof is 3.9 m, and its RQD is 79%. It is not cemented well with the upper sandstone and breaks and falls easily. The immediate roof is fine sandstone with an average thickness of 40.8 m, and its RQD is 89%; it is dense, hard and strong. According to drilling geological exploration results, the comprehensive coal–rock histogram of the 23,908 working face is shown in Figure 2. The uniaxial compressive strength of 9# coal is 14.82 MPa. The coal seam and floor of the 23,908 working face have weak bursting liability, and the roof has strong bursting liability.

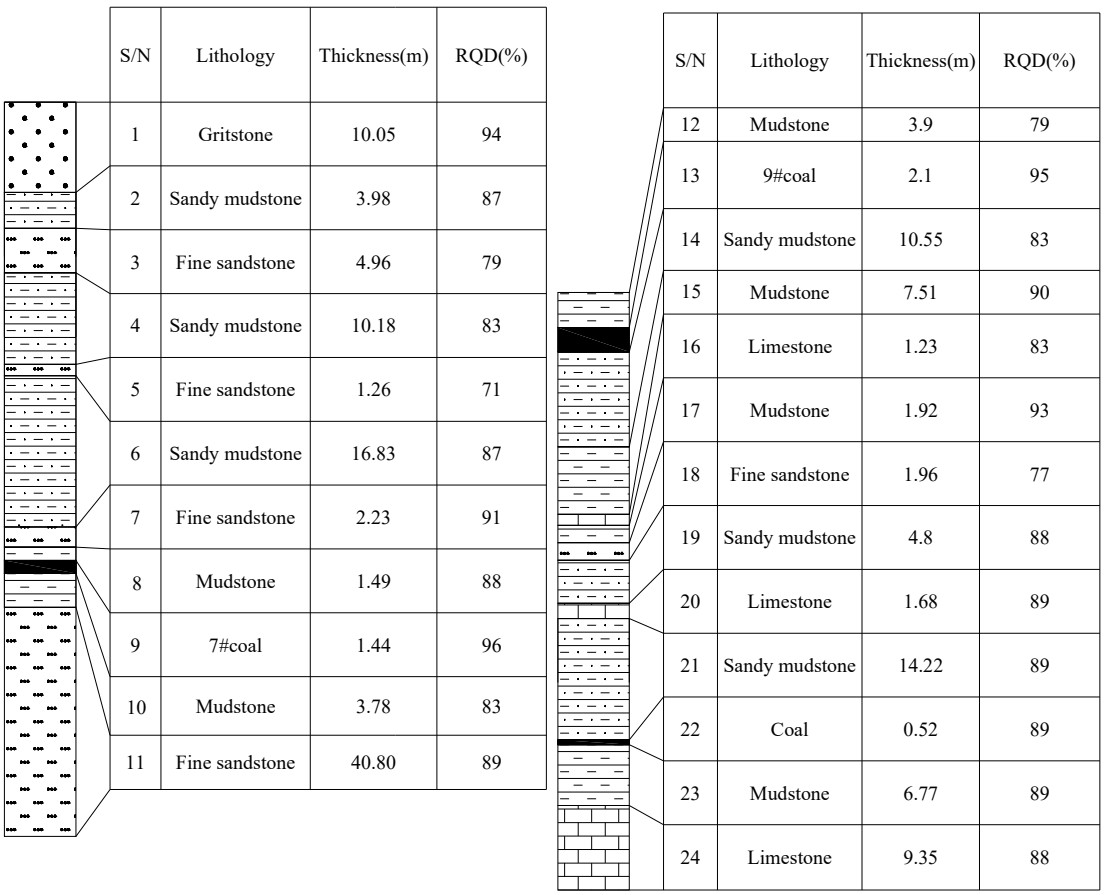

**Figure 2.** Comprehensive histogram of roof and floor of coal seam in 23,908 working face.

Based on a large number of practices, the empirical formula for calculating the height of the fracture zone is obtained [44], as shown in Table 2.

**Table 2.** Empirical formula of fracture zone height ($\Sigma M$ is cumulative mining height) [44].

| Lithology of Overlying Rock (UCS)/MPa | Computing Formula |
| --- | --- |
| Hard rock (40~80) | $H_{\mathrm{L}} = \frac{100\sum M}{1.2\sum M+2.0} \pm 8.9$ |
| Medium hard rock (20~40) | $H_{\mathrm{L}} = \frac{100\sum M}{1.6\sum M+3.6} \pm 5.6$ |
| Weak rock (10~20) | $H_{\mathrm{L}} = \frac{100\sum M}{3.1\sum M+5.0} \pm 4.0$ |
| Extremely weak rock (<10) | $H_{\mathrm{L}} = \frac{100\sum M}{5.0\sum M+8.0} \pm 3.0$ |

The immediate roof of the 23,908 working face is dense and hard fine sandstone, mainly composed of quartz and feldspar. The UCS of fine sandstone reaches 52 MPa, and thus it is classified as hard rock. According to the empirical formula of roof fracture zone height in Table 2, the height of the fracture zone is calculated as 37.6~55.4 m. Here, the

upper limit is taken to evaluate KES of overlying strata in the maximum range. According to a comprehensive histogram of the roof and floor of the coal seam in the 23,908 working face, the strata in the fracture zone, from bottom to top, are mudstone, fine sandstone, mudstone, mudstone, fine sandstone and sandy mudstone. The calculation results for the physical and mechanical parameters and vibrational wave energy of each rock stratum are shown in Table 3 (since the mudstone pseudo-roof is locally developed and ploughed during mining, it is not considered as the calculation object). It shows that the elastic energy released by fine sandstone I in the fracture zone is the largest during its initial fracture. The vibrational wave energy transmitting to the coal seam reaches $1.05 \times 10^5$ J, and the energy release coefficient $k_c$ is 10.47. Therefore, the immediate roof of fine sandstone is KES.

**Table 3.** Calculation results of seismic wave energy during initial fracture of rock strata in fractured zone.

| Rock Strata | Fine Sandstone I | Mudstone II | Mudstone III | Fine Sandstone II | Sandy Mudstone |
|---|---|---|---|---|---|
| $h$/m | 40.80 | 3.80 | 1.50 | 2.20 | 16.80 |
| $R_T$/MPa | 5.86 | 2.00 | 2.00 | 5.86 | 2.46 |
| $E_r$/GPa | 9.70 | 4.50 | 4.50 | 9.70 | 5.46 |
| $U_d$/J | $6.27 \times 10^7$ | $7.97 \times 10^4$ | $1.24 \times 10^4$ | $1.82 \times 10^5$ | $2.16 \times 10^6$ |
| $\eta$ | $5.5714 \times 10^{-4}$ | $1.2046 \times 10^{-3}$ | $1.3968 \times 10^{-3}$ | $1.2046 \times 10^{-3}$ | $9.1698 \times 10^{-4}$ |
| $U_{dh}$/J | $1.05 \times 10^5$ | $3.47 \times 10^1$ | $4.51 \times 10^0$ | $6.21 \times 10^1$ | $5.30 \times 10^2$ |
| $k_r$ | 10.47 | 0 | 0 | 0 | 0.05 |

## 3. Numerical Simulation Analysis of Mining Speed Effect on Energy Accumulation of Coal and Energy Release of KES

Different mining speeds will cause significant differences in stress transfer adequacy of the rock surrounding the working face. A higher mining speed leads to less sufficient stress adjustment in the surrounding rock. This section adopts FLAC3D simulation to study the effect of mining speed on the energy release of KES and energy accumulation of coal.

### 3.1. Numerical Simulation Methods

A plane strain model along the recovery direction of the 23,908 working face is established by FLAC3D. The constitutive models available in FLAC3D include the Mohr–Coulomb model, strain-softening model, strain-hardening model and creep model. The most commonly used constitutive model is the Mohr–Coulomb constitutive model. In the elastic stage of rock deformation under load, the Mohr–Coulomb constitutive model is exactly the same as the strain-softening constitutive model. The difference is that in the strain-softening constitutive model, when coal and rock mass reaches peak strength, rock strength decreases rapidly, and the mechanical properties of material deteriorate. Parameters such as cohesion, tensile strength and internal friction angle decrease rapidly with increasing plastic strain, which reflects the actual mechanical response of coal and rock mass under load. Therefore, the strain-softening constitutive model is selected in this model. Based on the results of the uniaxial compression test, uniaxial tensile test and shear test, the relevant physical and mechanical parameters of rock and coal mass can be obtained, as shown in Table 4.

The mechanical properties of fractured rock mass in the caving zone are simulated by the double-yield model. The relevant physical and mechanical parameters are shown in Table 5. The height of the caving zone is calculated by the statistical regression formula based on a large number of measurement data [45], namely

$$H_c = \frac{100h_c}{c_1 h_c + c_2} \tag{6}$$

where $h_c$ is mining height, m; $c_1$ and $c_2$ are parameters related to roof lithology, as shown in Table 6. The height of the caving zone is calculated to be 10 m.

**Table 4.** Physical and mechanical parameters of rock and coal strata.

| Rock Strata | Elastic Modulus/GPa | Poisson Ratio | Tensile Strength/MPa | Cohesion/MPa | Internal Friction Angle/° | Density /kg·m$^{-3}$ | Residual Tensile Strength/MPa | Residual Cohesion/MPa |
|---|---|---|---|---|---|---|---|---|
| Sandy mudstone | 5.46 | 0.22 | 2.46 | 2.00 | 32 | 2200 | 0.12 | 0.10 |
| Gritstone | 10.00 | 0.20 | 2.50 | 2.00 | 35 | 2600 | 0.13 | 0.10 |
| Mudstone | 4.50 | 0.28 | 2.00 | 1.20 | 27 | 2700 | 0.10 | 0.06 |
| Fine sandstone | 9.70 | 0.17 | 5.86 | 3.00 | 38 | 2800 | 0.29 | 0.15 |
| Coal | 1.19 | 0.36 | 0.50 | 0.80 | 23 | 1400 | 0.03 | 0.04 |

**Table 5.** Physical and mechanical parameters of double-yield model [45].

| Density/kg·m$^{-3}$ | Bulk Modulus/GPa | Shear Modulus/GPa | Internal Friction Angle/° | Dilation Angle/° |
|---|---|---|---|---|
| 1700 | 9.58 | 5.32 | 30 | 10 |

**Table 6.** Height coefficient of caving zone [45].

| Type of Immediate Roof | UCS/MPa | Coefficient | |
|---|---|---|---|
| | | $c_1$ | $c_2$ |
| Hard | >40 | 2.1 | 16 |
| Medium–hard | 20~40 | 4.7 | 19 |
| Weak | <20 | 6.2 | 32 |

According to ground stress testing results, the initial ground stress in the model is set as follows: $\sigma_{xx}$ = 26.11 MPa, $\sigma_{yy}$ = 30.71 MPa, $\sigma_{zz}$ = 27.25 MPa. The vertical displacement is fixed at the bottom of the model and the horizontal displacement is fixed at both sides. The vertical stress at the top of model is set as 25.8 MPa, which represents the overlying strata load, as shown in Figure 3.

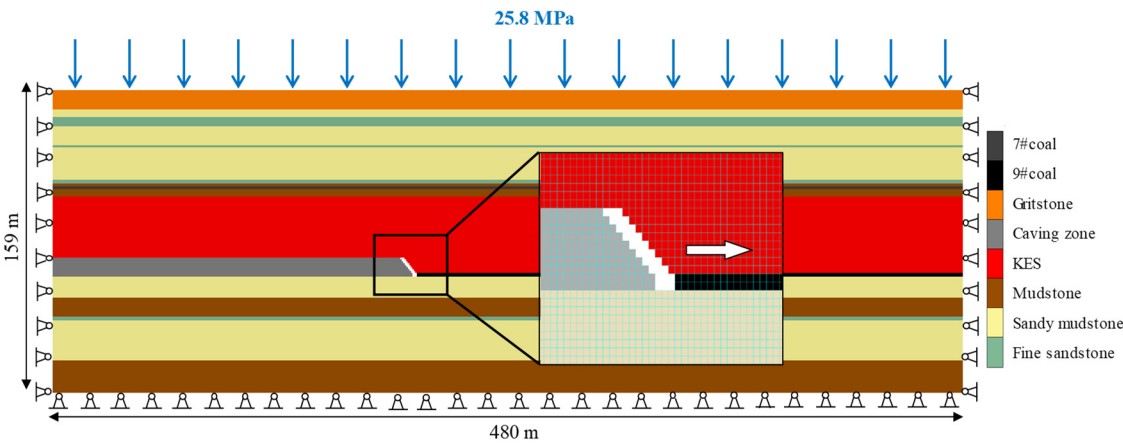

**Figure 3.** FLAC3D numerical model.

According to existing research results, a mining speed of 4.0 m/d is usually used as safe mining speed or medium mining speed. Therefore, in this paper, mining speed is divided into three regional ranges: low speed (<4.0 m/d), medium speed (4.0 m/d~6.0 m/d) and high speed (>6.0 m/d). Considering that the shearer cutting depth in the 23,908 working face is 0.6 m, to ensure that numerical simulation schemes match actual production conditions, 3.6 m/d, 4.8 m/d and 6.0 m/d are selected as low, medium and high mining speeds, respectively, in the numerical simulation model. In order to facilitate grid elements'

excavation in numerical simulation, the coal mass is divided into elements with a 0.6 m grid length along the *x* direction. In order to simulate the whole process from initial mining to the initial weighting stage of the 23,908 working face, considering that the slope length of the working face is 193.7 m, the whole simulation process starts from the left side of the model and ends at 192 m with different mining speeds.

In the model, elastic strain energy accumulation in each coal mass element is calculated by Equation (7). Considering that most of the elastic strain energy released by rock strata fracture is dissipated as heat and acoustic emissions, seismic efficiency $\Omega$ only reaches 0.26%~3.6% [46]; therefore, the elastic strain energy released by each element in KES in the model is calculated by Equation (8), as follows

$$U_{si} = V \cdot \left[ \sigma_1^2 + \sigma_2^2 + \sigma_3^2 - 2\nu(\sigma_1\sigma_2 + \sigma_2\sigma_3 + \sigma_1\sigma_3) \right]/2E \tag{7}$$

$$U_{di} = \Omega V \left[ \sigma_1^2 + \sigma_2^2 + \sigma_3^2 - 2\nu(\sigma_1\sigma_2 + \sigma_2\sigma_3 + \sigma_1\sigma_3) \right]/2E \tag{8}$$

where *V* is unit volume, m$^3$; $\sigma_1$, $\sigma_2$ and $\sigma_3$ are three spatial principal stresses, MPa; $\nu$ is Poisson's ratio of coal and rock mass; *E* is the elastic modulus of coal and rock mass, GPa; $\Omega$ is seismic efficiency. The value used in this paper is 3.6%.

*3.2. Results Analysis*

3.2.1. Evolution of Peak Vertical Stress in Coal

The evolution process of peak vertical stress of coal under different mining speeds is shown in Figure 4. It indicates that peak vertical stress increases rapidly at first, reaches a maximum value and then decreases slowly and tends towards a constant value. The higher the mining speed is, the more rapidly the peak vertical stress increases and the greater the working face advancing distance when peak vertical stress reaches the maximum value. When the mining speed is 3.6 m/d and the working face advancing distance is 61 m, the maximum vertical stress is 54.3 MPa and the stress concentration factor is 2.0. When the mining speed is 4.8 m/d and the working face advancing distance is 77 m, the maximum vertical stress reaches 59.8 MPa and the stress concentration coefficient is 2.2. When the mining speed is 6.0 m/d and the working face advancing distance is 90 m, the maximum vertical stress is 61.1 MPa and the stress concentration factor is 2.3. The above results show that with increasing mining speed, the initial weighting step of the working face increases, resulting in an increase in front abutment pressure in coal mass when weighting occurs.

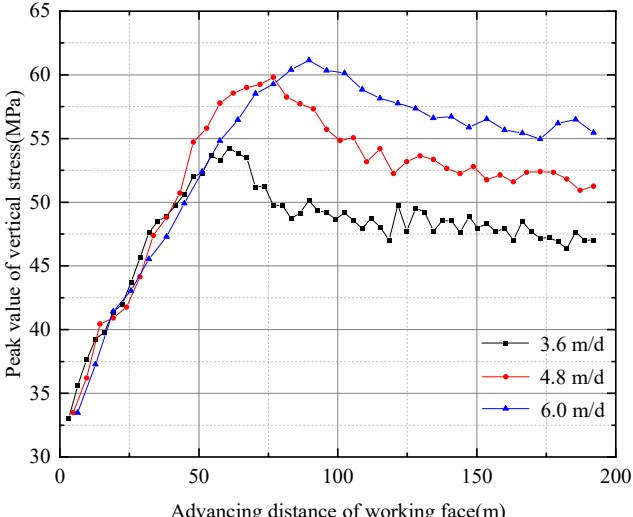

**Figure 4.** Evolution of peak vertical stress in coal with different mining speeds.

### 3.2.2. Distribution Characteristics of Vertical Stress

Figure 5 shows the vertical stress distribution in the surrounding rock under different mining speeds when the excavation distance of the working face reaches 192 m. It demonstrates that the increased vertical stress zone ahead of the working face presents relative "shield" distribution. With increasing mining speed, the range of high-stress areas in the fracture zone increases gradually, and the range of low-stress areas decreases gradually. At the same time, the peak value of vertical stress increases gradually. When the mining speed is 3.6 m/d, the peak value of vertical stress is about 53.3 MPa. When the mining speed is 4.8 m/d, the peak value of vertical stress is about 55.7 MPa, which is 4.5% greater than the former. When the mining speed is 6.4 m/d, the peak value of vertical stress is about 60.6 MPa, which is 8.8% higher than the former. Meanwhile, Figure 6 shows that the range of the vertical stress concentration zone increases with the increase in mining speed, and the peak value of vertical stress gradually approaches the free surface of the mining space. The above vertical stress field distribution characteristics are consistent with existing research [47], which means the numerical model in this paper is feasible.

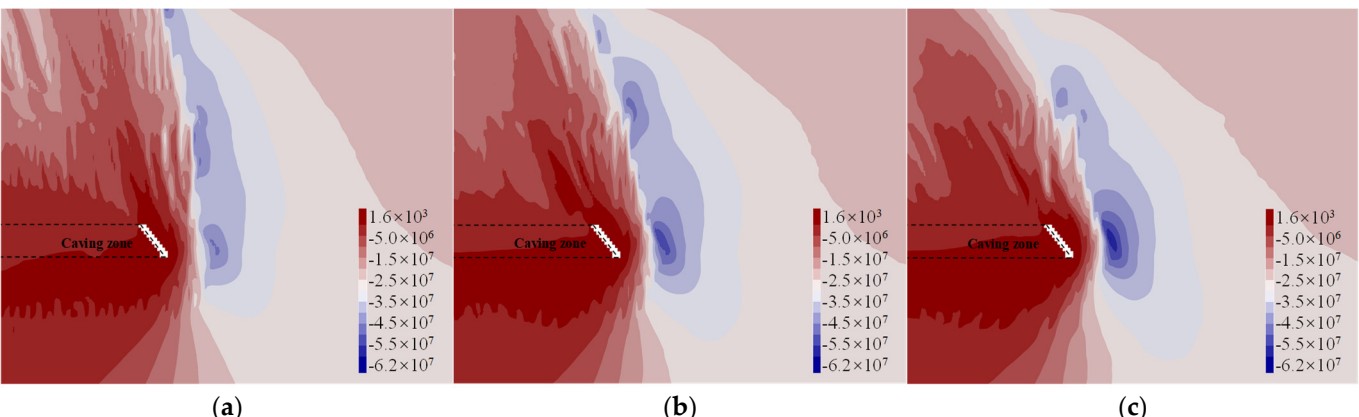

**Figure 5.** Vertical stress distribution under different mining speeds (when advancing distance of working face is 192 m, unit: Pa): (**a**) mining speed 3.6 m/d; (**b**) mining speed 4.8 m/d; (**c**) mining speed 6.0 m/d.

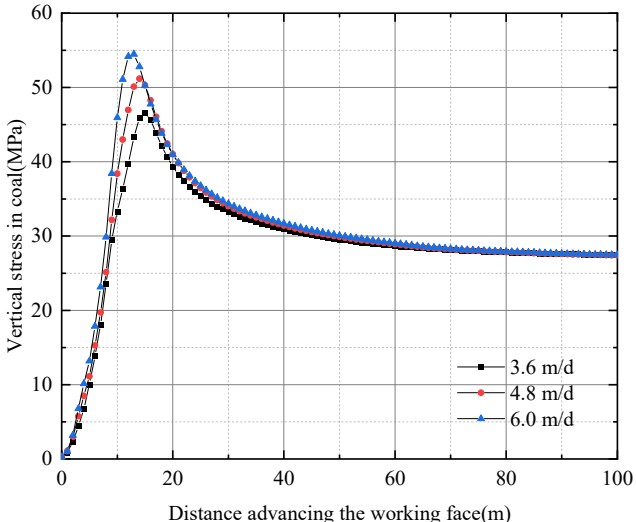

**Figure 6.** Vertical stress in coal mass under different mining speed conditions (when advancing distance of working face is 192 m).

### 3.2.3. Evolution of Peak Elastic Strain Energy in Coal

Figure 7 shows the evolution process of peak elastic strain energy of coal under different mining speeds. It indicates that the peak value of elastic strain energy increases rapidly at first, reaches maximum value and then decreases slowly and tends toward a constant value. The higher the mining speed is, the greater the growth amplitude of peak elastic strain energy and the greater the advancing distance of the working face when the peak elastic strain energy reaches the maximum value. When the mining speed is 3.6 m/d and the working face advancing distance is 61 m, the peak elastic strain energy reaches a maximum value of $2.08 \times 10^6$ J. When the mining speed is 4.8 m/d and the working face advancing distance is 77 m, the peak elastic strain energy reaches a maximum value of $2.39 \times 10^6$ J. When the mining speed is 6.0 m/d and the working face advancing distance is 90 m, the peak elastic strain energy reaches a maximum value of $2.48 \times 10^6$ J. The above results are consistent with the evolution process of peak vertical stress. It is observed that with increasing mining speed, the initial weighting step of the working face increases, resulting in an increase in elastic strain energy accumulation in coal when weighting occurs.

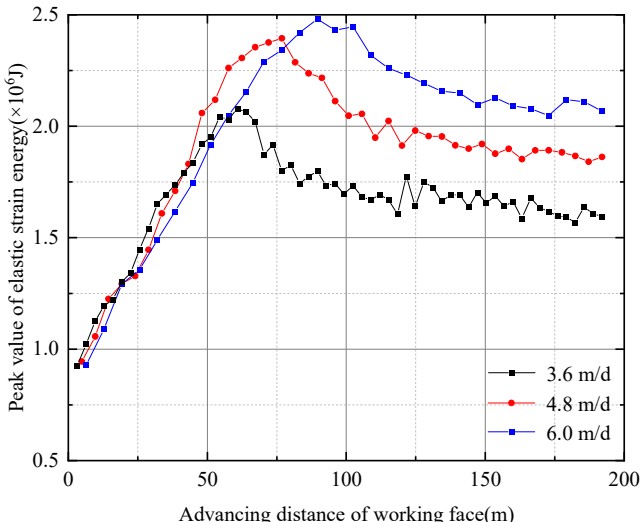

**Figure 7.** Evolution of peak elastic strain energy in coal with different mining speeds.

### 3.2.4. Distribution Characteristics of Elastic Strain Energy

Figure 8 shows the elastic strain energy distribution with different mining speeds when the working face excavation distance reaches 192 m. It indicates that the coal seam is the main layer of elastic energy accumulation. With increasing mining speed, the peak value of elastic energy accumulation in the coal gradually increases. When the mining speed is 3.6 m/d, the peak value of elastic energy accumulation is about $1.99 \times 10^6$ J. When the mining speed is 4.8 m/d, the peak value of elastic energy accumulation is about $2.33 \times 10^6$ J, which is 17.1% higher than the former. When the mining speed is 6.0 m/d, the peak value of elastic energy accumulation is about $2.58 \times 10^6$ J, which is 10.7% higher than the former. At the same time, it can be seen in Figure 9 that the accumulation range of elastic strain energy increases with increasing mining speed and the maximum elastic strain energy accumulation zone gradually approaches the free surface of mining space. Therefore, from the perspective of energy accumulation, it can be concluded that the increase in mining speed contributes to the increase in storage range and storage level of static elastic strain energy in coal, which is one of the most significant reasons for the increase in rock burst risk at the working face.

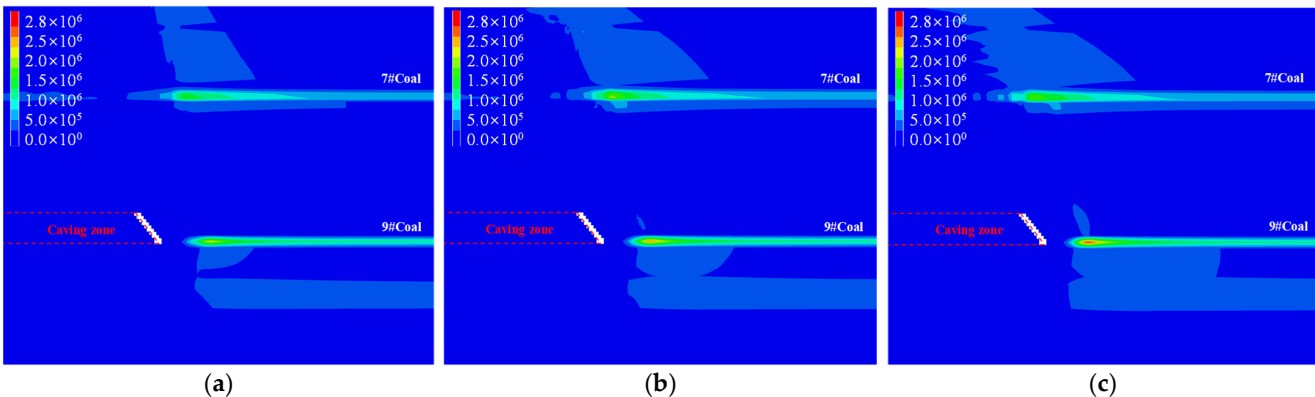

**Figure 8.** Elastic strain energy distribution of coal and rock mass under different mining speeds (when advancing distance of working face is 192 m, unit: J): (**a**) mining speed 3.6 m/d; (**b**) mining speed 4.8 m/d; (**c**) mining speed 6.0 m/d.

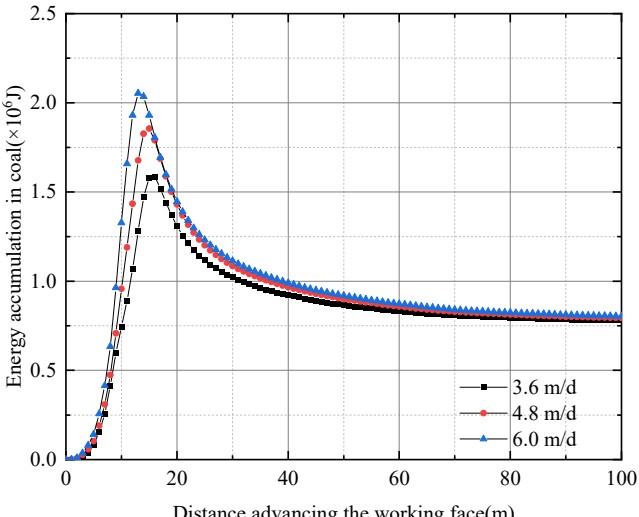

**Figure 9.** Elastic strain energy distribution in coal under different mining speeds (when advancing distance of working face is 192 m).

3.2.5. Evolution Characteristics of Energy Release in KES

On the basis of numerical simulation results, the distribution of elastic strain energy release elements in KES under different mining speeds is counted and the results are shown in Figure 10. It indicates that when the mining speed is 3.6 m/d, the energy release unit is mainly distributed at 0~25 m above the roof, the energy release level is in the range of $10^3 \sim 10^4$ J, and the maximum energy release is 9,930 J. When the mining speed increases to 4.8 m/d, the distribution range of energy release elements further expands to 0~30 m above the coal seam roof, and high-level energy release elements of $10^4 \sim 10^5$ J appear in KES, and the maximum energy release increases to 12,019 J. When the mining speed further increases to 6.0 m/d, the energy release elements almost cover the whole horizon of KES, the number of high-level energy release elements increases slightly, and the maximum energy release is 11,616 J. The above results show that with increasing mining speed, the energy level and distribution range of mining seismicities show obvious transition growth characteristics. Mining earthquakes above $10^4$ J are more likely to occur when the mining speed is greater than 3.6 m/d, which is another significant reason for the increase in rock burst risk at the working face.

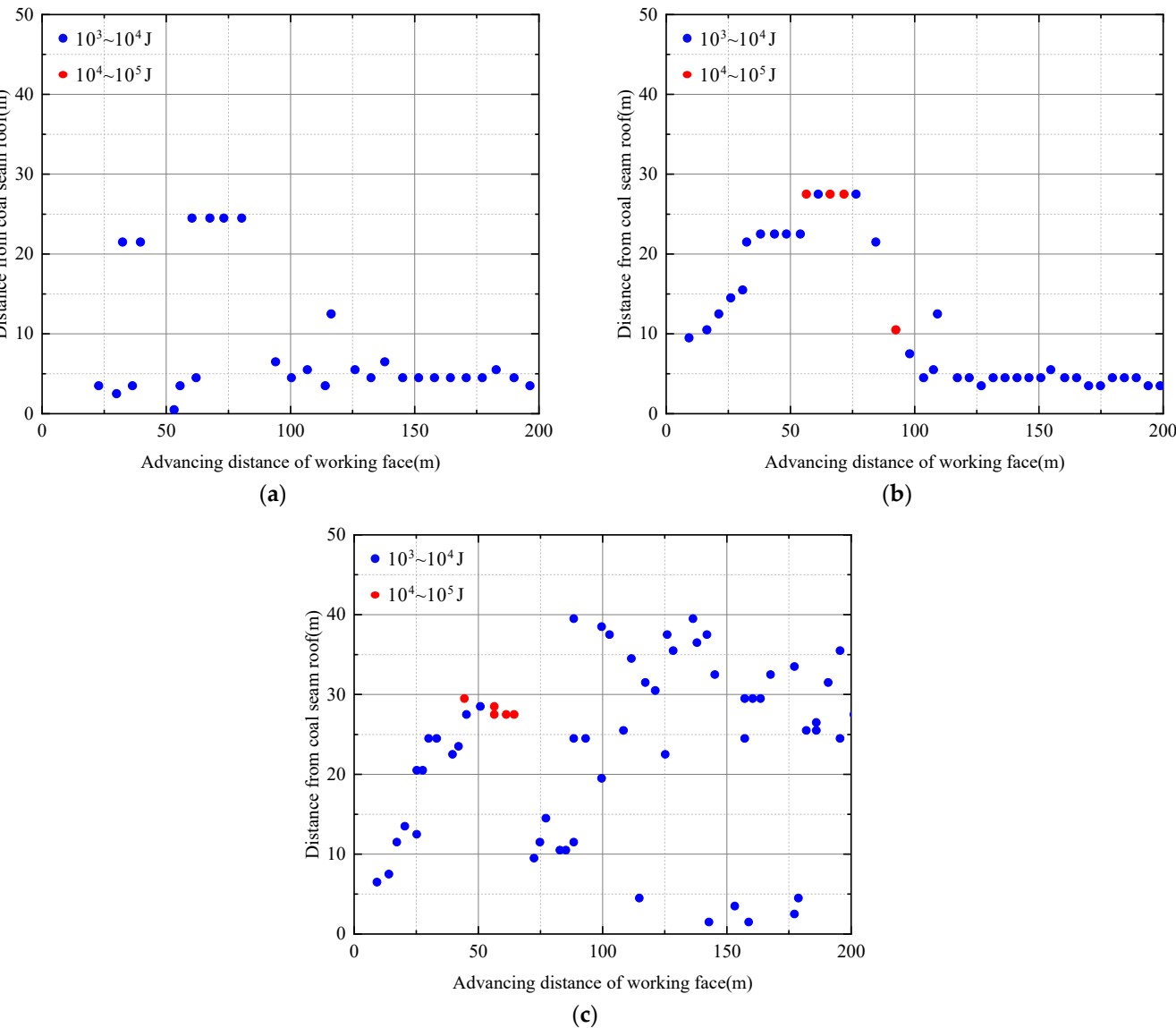

**Figure 10.** Evolution process of energy release in KES under different mining speeds: (**a**) mining speed 3.6 m/d; (**b**) mining speed 4.8 m/d; (**c**) mining speed 6.0 m/d.

## 4. Discussion

### 4.1. Rock Burst Risk Assessment Considering Mining Speed Effect

In order to quantitatively assess rock burst risk considering the mining speed effect, it is necessary to integrate the energy accumulation degree in coal mass and the energy release degree of KES under different mining speed conditions. At the same time, the attenuation process of vibrational wave energy induced by KES fracture needs to be considered. Based on the minimum energy principle of rock mass dynamic failure and the dynamic and static load superposition principle of rock burst, a rock burst risk evaluation index considering the influence of the mining speed effect is established in this section.

Rock burst is a dynamic phenomenon occurring when the mechanical equilibrium state of the rock surrounding coal is broken, and the released energy is greater than the energy consumed during the coal mass failure process [48]. When rock burst occurs, the minimum consumed energy is

$$U_{\text{bmin}} = \frac{\sigma_{\text{bmin}}^2}{2E} \tag{9}$$

where $\sigma_{\text{bmin}}$ is a critical stress of rock burst.

According to reference [48], the occurrence of rock burst meets the following conditions

$$\sigma_\mathrm{j} + \sigma_\mathrm{d} \geq \sigma_\mathrm{bmin} \tag{10}$$

where $\sigma_\mathrm{j}$ is the static load in coal and rock mass, and $\sigma_\mathrm{d}$ is the dynamic load induced by mine tremors in coal and rock mass.

The experimental study [49] shows that when the uniaxial compressive strength $R_\mathrm{c}$ of coal is greater than 20 MPa, the critical dynamic load of dynamic failure of coal is about 50 MPa. When the uniaxial compressive strength $R_\mathrm{c}$ of coal is less than 16 MPa, the critical dynamic load of dynamic failure of coal is about 70 MPa. When the uniaxial compressive strength $R_\mathrm{c}$ of coal is between 16 and 20 MPa, the critical dynamic load of dynamic failure is about 50 to 70 MPa. In this work, 9# coal was shown to have a uniaxial compressive strength test of 14.82 MPa and an elastic modulus of 1.19 GPa. Substituting these into the above equation, the minimum energy $U_\mathrm{bmin}$ of dynamic failure of coal in this case is $4.1176 \times 10^6$ J. In order to assess rock burst risk under different mining speeds, a new energy index was established, as follows:

$$\phi_{vi} = \frac{U_{vi}}{U_\mathrm{bmin}} \tag{11}$$

where $U_{vi}$ is the maximum value of elastic strain energy in coal under the action of dynamic and static load superposition with different mining speeds, which is obtained by scalar superposition of the peak elastic strain energy accumulated in coal and peak elastic strain energy released by KES, as shown in Equation (12).

$$U_{vi} = U_{si} + U_{di} \cdot H^{-2} e^{-\eta H} \tag{12}$$

Table 7 shows the corresponding relationship between $\Phi_{vi}$ and rock burst risk.

**Table 7.** Relationship between energy index and rock burst risk level.

| $\Phi_{vi}$ | Rock Burst Risk Level |
|---|---|
| $0 < \Phi_{vi} \leq 0.25$ | None |
| $0.25 < \Phi_{vi} \leq 0.5$ | Weak |
| $0.5 < \Phi_{vi} \leq 0.75$ | Medium |
| $0.75 < \Phi_{vi} \leq 1$ | Strong |

Equations (11) and (12) are used to calculate the energy index of different advancing distances of the working face under different mining speeds, as shown in Figure 11. With the working face advancing, KES reaches the initial fracture state at first, leading to the elastic strain energy and vertical stress in coal mass increasing. After the initial fracture state, the vertical stress and elastic strain energy in coal mass significantly decrease. Under deep coal mining conditions, rock burst usually occurs in a high-stress environment with low dynamic load disturbance and a high static load. Energy accumulation in coal mass is the basis for the inoculation and development of rock burst, and the energy released by KES plays an induction role. As a result, the energy index under different mining speeds first increases and then decreases during mining of the working face, and finally tends to be constant, having strong consistency and synchronization with the transformation of vertical stress and elastic strain energy in coal mass.

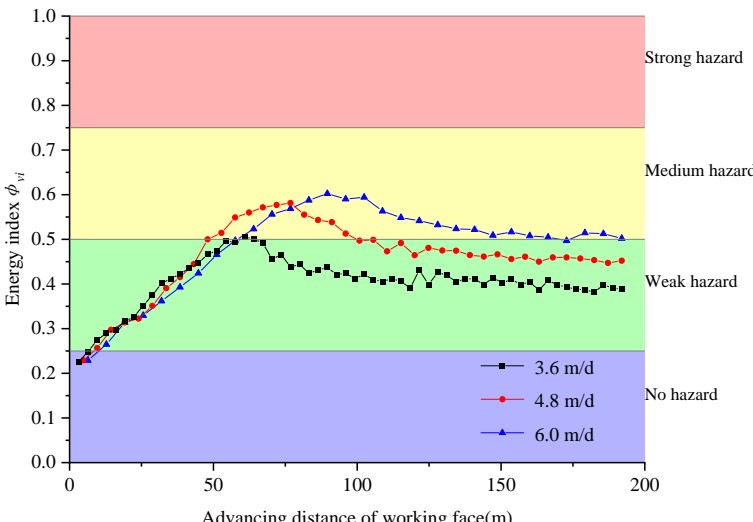

**Figure 11.** The change in energy index during mining process with different mining speeds.

The evolution process of rock burst risk under different mining speeds experiences three stages, namely no rock burst risk, weak rock burst risk and medium rock burst risk. With increasing mining speed, the proportion of medium rock burst risk area increases gradually. When the mining speed is 3.6 m/d, the proportion of medium rock burst risk area is only 6.7%. The working face undergoes a weak rock burst risk stage at the middle and later mining stages. When the mining speed is 4.8 m/d, the proportion of medium rock burst risk area increases to 30.0%, but the working face undergoes a weak rock burst risk stage at the middle and later mining stages. When the mining speed is 6.0 m/d, the proportion of medium rock burst risk area increases to 70.0% and the working face basically undergoes a medium rock burst risk stage at the middle and later mining stages.

To ensure that the rock burst risk is below the weak state and maximizes mining efficiency of the working face, the mining speed should be set as 3.6 m/d. It is worth noting that this work only analyzes the evolution process of rock burst risk from the perspective of the mining speed of the working face, which only considers the influence of mining-induced stress. However, during the actual production process, tectonic stress also plays an important role in rock burst risk. Therefore, during the actual production process, in addition to selecting a reasonable mining speed, it is also important to strengthen the pressure relief of coal mass at the advanced mining disturbance area and geological structure area so as to reduce energy accumulation in coal mass. At the same time, reasonable roof cutting and pressure relief measures need to be implemented for the KES to reduce elastic strain energy release when KES breaks so as to minimize rock burst risk.

### 4.2. Engineering Practice Test

To verify the rationality and validity of the KES obtained by theoretical analysis, taking 10 days as a time window, the distribution of microseismic events of the 23,908 working face at different mining stages is drawn in Figure 12. In stages (a), (b) and (c), the mining speed of the working face is 1.2 m/d, 2.4 m/d and 3.6 m/d, respectively. It can be seen from the microseismic distribution plane in Figure 12 that microseismic events (especially those in the range of $10^3 \sim 10^4$ J) are concentrated in the advanced area of two roadways at the working face, indicating that the rock burst risk in this area is relatively high. Meanwhile, Figure 12 shows that microseismic events (especially those in the range of $10^3 \sim 10^4$ J) are concentrated in the zone of 0~40 m above the coal seam roof, which is consistent with the occurrence range of thick, hard fine sandstone roofs, indicating that microseismic events are mainly induced by fine sandstone roof fracture, which verifies the correctness of the conclusion that the fine sandstone roof is KES, as posited in the theoretical analysis in Section 2.

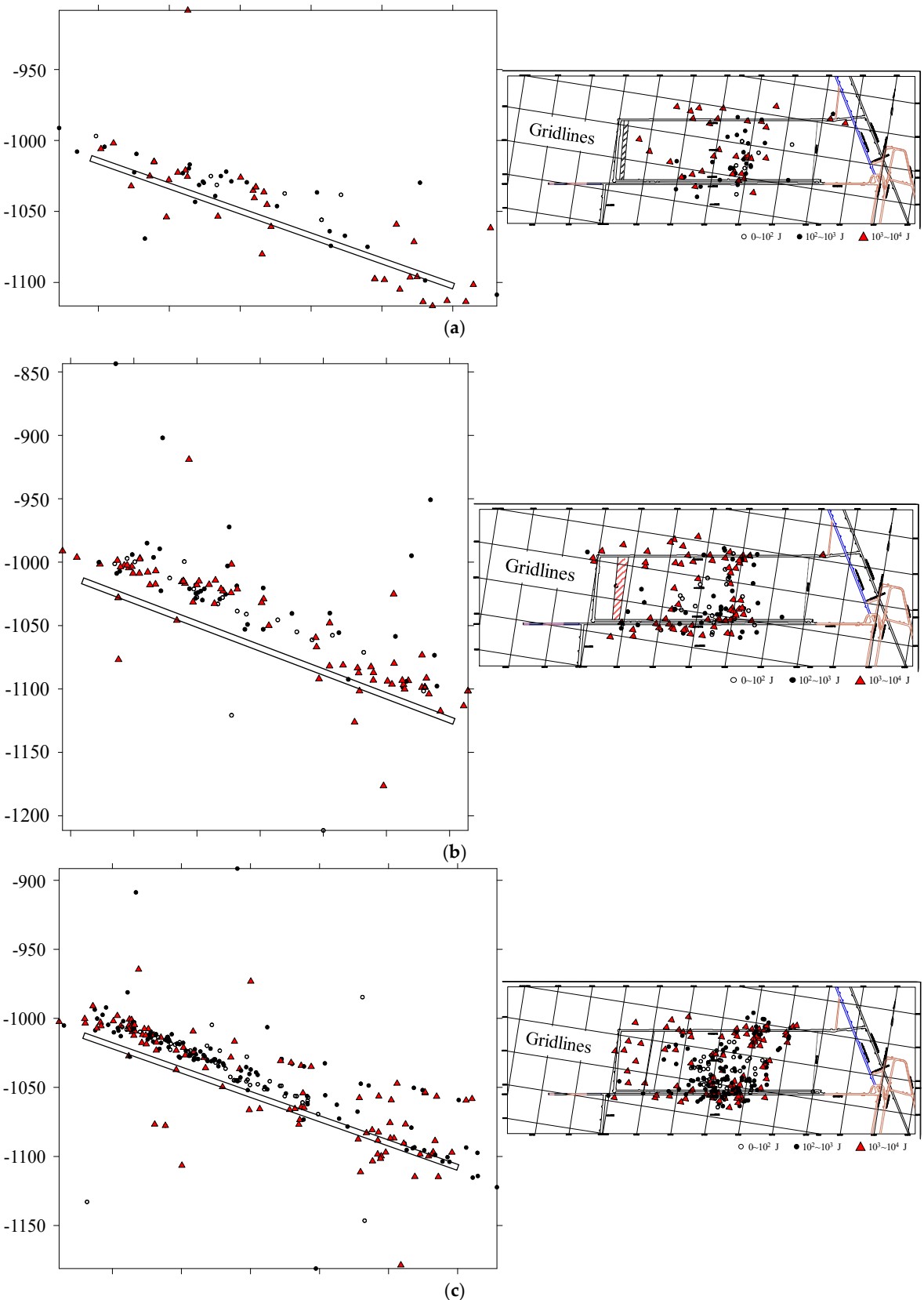

**Figure 12.** Microseismic distribution of 23,908 working face in different mining stages (shadowed part is the mining range of working face in corresponding state): (**a**) microseismic distribution from 7 March to 17 March; (**b**) microseismic distribution from April 3 to April 13; (**c**) microseismic distribution from 20 April to 30 April.

To verify the rationality and validity of vertical stress distribution in coal mass obtained by numerical calculation, taking a mining speed of 3.6 m/d as an example, a comparison between the field testing result and numerical simulation result is shown in Figure 13. It can be concluded that, both in trend and precision, the numerical simulation result well reflects vertical stress distribution in the field, validating the rationality of our numerical model.

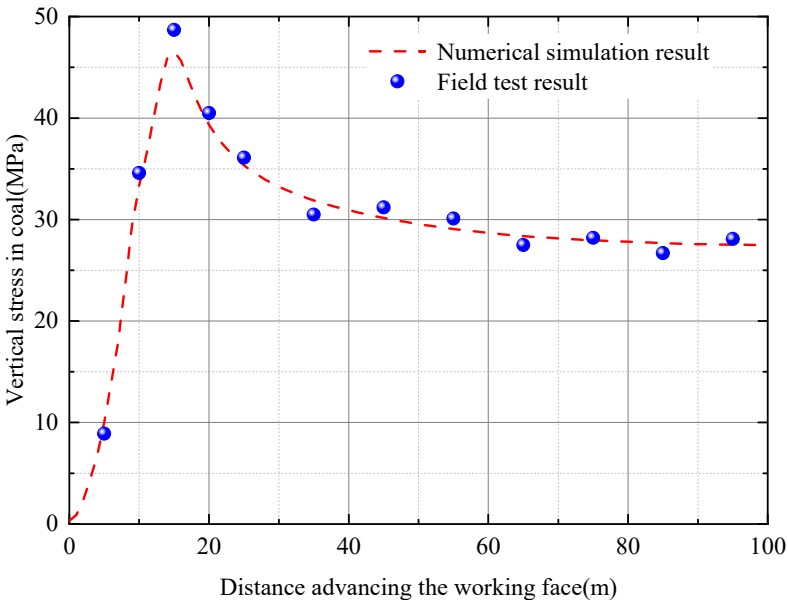

**Figure 13.** Comparison of vertical stress in coal mass with mining speed of 3.6 m/d.

In order to verify the rationality of the safe mining speed obtained by numerical calculation, taking 10 days as time window, the cumulative microseismic energy and the number of cumulative microseismic events in each energy interval are counted, as shown in Figure 14. In stage (a), (b) and (c), the mining speed of the working face is 1.2 m/d, 2.4 m/d and 3.6 m/d, respectively. It can be seen from Figure 14 that microseismic energy levels are mainly concentrated in the range of $10^2 \sim 10^3$ J and $10^3 \sim 10^4$ J under different mining speeds. When the mining speed is 1.2 m/d, the number of microseismic events in $10^2 \sim 10^3$ J and $10^3 \sim 10^4$ J levels accounts for 37% and 52% of the total. When the mining speed is 2.4 m/d, the number of microseismic events in $10^2 \sim 10^3$ J and $10^3 \sim 10^4$ J levels accounts for 31% and 52% of the total. When the mining speed is 3.6 m/d, the number of microseismic events in $10^2 \sim 10^3$ J and $10^3 \sim 10^4$ J levels accounts for 43% and 37% of the total. When the mining speed is 1.2 m/d, 2.4 m/d and 3.6 m/d, the maximum microseismic energy is 6880 J, 8680 J and 8627 J, respectively, which does not exceed the minimum energy ($1 \times 10^4$ J) inducing rock burst, indicating that rock burst risk can be effectively reduced and safe mining conditions for the 23,908 working face can be guaranteed when the mining speed is set as 3.6 m/d.

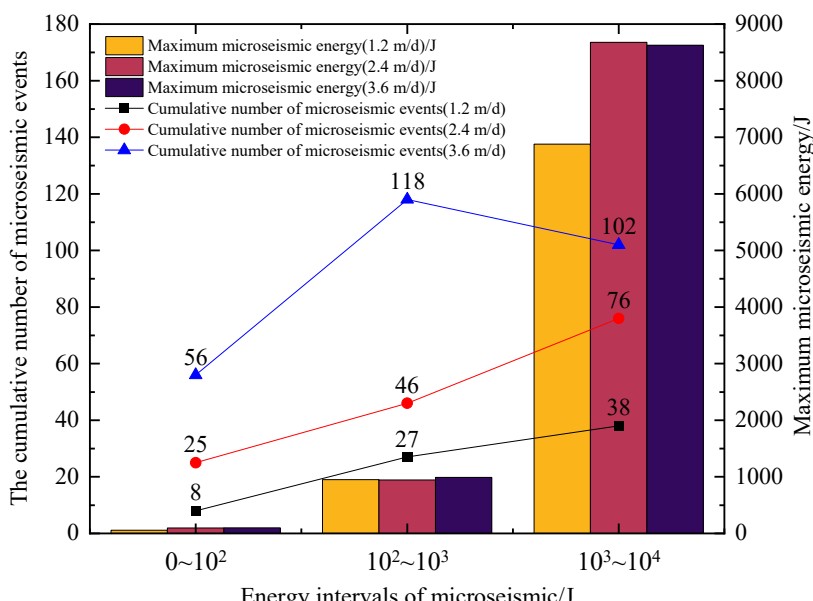

**Figure 14.** The cumulative microseismic energy and event number in each energy interval under different mining speeds.

## 5. Summaries and Conclusions

In this work, taking the 23,908 working face in the Zhangshuanglou Coal Mine in China as an engineering background, theoretical analysis, numerical simulation and field measurement are comprehensively conducted to investigate the mining speed effect on KES and the evaluation method for rock burst risk. The following conclusions can be drawn from the above results:

(1) The principle of KES for roof-type rock burst was proposed. The ratio of energy released by initial fracture of rock strata in the fracture zone to the minimum energy inducing rock burst was defined as the energy release coefficient. Combined with the geological conditions of the 23,908 working face in Zhangshuanglou Coal Mine, it was determined that the KES in overlying strata was a fine sandstone roof with a thickness of 40.8 m.

(2) The mechanism of increasing rock burst risk caused by increasing mining speed can be explained from two perspectives. On the one hand, an increase in mining speed leads to an increase in the storage range and energy level of static elastic strain energy in coal mass. On the other hand, an increase in mining speed leads to an increase in the energy level and distribution range of mining seismicities induced by fracture of KES. The rock burst risk of the working face increases under the combined influence of the above two factors.

(3) Based on the minimum energy principle of rock mass dynamic failure and the dynamic and static load superposition principle of rock burst, an evaluation index $\Phi_{vi}$ of rock burst risk considering the influence of mining speed was established, and it was divided into four levels. Accordingly, it was determined the 23,908 working face basically undergoes a weak rock burst risk stage or below when the mining speed is 3.6 m/d.

(4) The engineering practice results show that when the mining speed is 3.6 m/d, the maximum microseismic energy is 8627 J, which does not exceed the minimum energy $(1 \times 10^4$ J) inducing rock burst. Therefore, rock burst risk can be effectively reduced and safe mining conditions for the 23,908 working face can be guaranteed when the mining speed is set as 3.6 m/d.

**Author Contributions:** Conceptualization, W.L. and S.T.; methodology, H.T.; software, Y.L. and L.T.; validation, X.L., K.M. and H.Z.; formal analysis, W.L.; writing—original draft preparation, W.L.;



writing—review and editing, S.T. and J.M.; project administration, S.T. All authors have read and agreed to the published version of the manuscript.

**Funding:** The work in this paper was financially supported by the National Natural Science Foundation of China (grant number: 51874281) and the China National Natural Science Foundation Youth Funding Project (grant number: 52004270).

**Institutional Review Board Statement:** Not applicable.

**Informed Consent Statement:** Not applicable.

**Data Availability Statement:** Data sharing not applicable.

**Acknowledgments:** The authors would like to thank the engineering technicians at the Zhangshuanglou Coal Mine for their enthusiastic assistance and suggestions.

**Conflicts of Interest:** The authors declare no conflict of interest.

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
