# Peer review of "A New Method to Assess Thick, Hard Roof-Induced Rock Burst Risk Based on Mining Speed Effect on Key Energy Strata"

_sustainability, doi:10.3390/su142215054_

Round 1

Reviewer 1 Report

I liked the manuscript very much, but when studying the manuscript, the following questions arose:

1. The "key strata inducing rock burst" method, can you write a method object with practical examples of its use [line 130]. The explanations in the article are not enough to understand the whole essence of this method.

2. "The coal seam and the floor of lava 23908 are weakly cracking, the roof is cracking heavily" [p. 192-193]. Fracturing of rocks is one of the main characteristics of the state of rocks, therefore, quantitative indicators such as RQD (core recovery quality index) should be used to assess the degree of fracture.

3. It is not entirely clear how the parameters of the physical and mechanical properties of rocks and coal were obtained? [Table 4, line 241].

4. It is not clear how, with a greater uniaxial compression of coal (20 MPa), the critical load under shock-dynamic impact is the smallest (50 MPa)? This is contrary to logic, rocks with greater strength are able to withstand a higher critical load and vice versa [lines 390-395].

Reviewer 2 Report

This manuscript outlines a new method to assess hard-thick roof induced rock burst risk based on the mining speed effect on key energy strata. The topic is interesting, yet several weaknesses appear in this manuscript, which can be seen from the reasons:

1) I found some obvious grammatical issues across the paper, please proofread the paper.

2) At time, the first-person narration is used in some sentences of this manuscript. A scientific paper needs to be written as the third-person narration.

3) The highlights are too long, which should be shorten significantly.

4) The introduction seems too long, which should be shorten significantly. The section introduction presents why you carried out the working in this manuscript.

5) In lines 130-167, the authors reviewed the literature but it is not useful for readers to understand the novelty of this paper I think. The authors need to provide insights of those studies and mention how those findings are correlated to the focus of your study.

6) In lines 171, the authors need to review the studies to show the key mechanism of the mining speed affecting engineering.

7) For Eqs. (1-5), and the latter similarly, did the authors derive those or adopt them from the literature? if the latter, you need to give references, otherwise give the derivation details.

8) The authors need to give references to Table 2.

9) How did you choose the parameters in Tables 4-6? It need to be clarified.

10) In lines 490-492, the authors need to explain why it increases first and then decreases in Fig. 11?

11) The authors need to do a direct comparison of the mining speed effect results from the field test and numerical simulation to verify the numerical analysis.

12) The conclusion needs to be concise, it is too long and lacks focus.

Reviewer 3 Report

Reviewer has the following points/comments to be covered before the publication of the work:

·         The novelty of this work needs to be elaborated more. Why this study is so important? How it differentiates from the other methods.

·         Add references for equations 1-5.

·         Add references for each formula in Table 2.

·         The numerical part using FLAC3D is not rigorous and is missing information that are crucial to the reproduction of the proposed numerical simulations

·         No comparison was made to verify the results.

Round 2

Reviewer 3 Report

Authors have addressed satisfactory all cThe authors have satisfactorily addressed most of my comments.